# Translocation-Related Sarcomas

**DOI:** 10.3390/ijms19123784

**Published:** 2018-11-28

**Authors:** Kenji Nakano, Shunji Takahashi

**Affiliations:** Department of Medical Oncology, Cancer Institute Hospital of the Japanese Foundation for Cancer Research, Tokyo 135-0063, Japan; s.takahashi-cheomotherapy@jfcr.or.jp

**Keywords:** Soft tissue sarcoma, chromosomal translocation, translocation-related sarcoma, Ewing sarcoma breakpoint region 1, forkhead box transcription factor O, transcription factor E3, anaplastic lymphoma kinase, neurotrophic tyrosine kinase

## Abstract

Chromosomal translocations are observed in approximately 20% of soft tissue sarcomas (STS). With the advances in pathological examination technology, the identification of translocations has enabled precise diagnoses and classifications of STS, and it has been suggested that the presence of and differences in translocations could be prognostic factors in some translocation-related sarcomas. Most of the translocations in STS were not regarded as targets of molecular therapies until recently. However, trabectedin, an alkylating agent, has shown clinical benefits against translocation-related sarcoma based on a modulation of the transcription of the tumor’s oncogenic fusion proteins. Many molecular-targeted drugs that are specific to translocations (e.g., anaplastic lymphoma kinase and tropomyosin kinase related fusion proteins) have emerged. The progress in gene technologies has allowed researchers to identify and even induce new translocations and fusion proteins, which might become targets of molecular-targeted therapies. In this review, we discuss the clinical significance of translocation-related sarcomas, including their diagnoses and targeted therapies.

## 1. Introduction

### 1.1. What Is a Chromosomal Translocation?

A chromosomal translocation is a chromosome abnormality characterized by chromosomal rearrangements that result in the fusion of genes, which were originally separate. These “fusion genes” are known to be the causes of various diseases, including malignant diseases [1]. The first fusion gene derived from a chromosomal translocation that was investigated both in tumor genesis and as a treatment target is *BCR-ABL*. This fusion gene, from the t(9;22)(q34;q11) translocation observed in chronic myeloid leukemia (CML), has been called the “Philadelphia chromosome”, and tyrosine kinase inhibitors targeting *BCR-ABL* produced robust responses and clinical benefits against CML, opening the era of molecular-targeting treatments in cancer therapy [2]. About 10 years after the emergence of *BCR-ABL* therapy targeting CML, a new driver mutation based on fusion genes derived from chromosomal translocations was identified in solid tumors, i.e., *EML4-ALK* from t(2;5)(p23;q35) translocation [3]. Anaplastic lymphoma kinase (ALK)-related translocations are observed in only 2%–5% of non-small cell lung cancers (NSCLCs), but molecular-targeted therapy to *ALK* showed highly significant responses that were the same as those observed when targeting *BCR-ABL* in CML, bringing about a new paradigm of medical research and precision medicine [4,5,6].

There are many chromosomal translocations related to and/or detected in specific malignant diseases. More than 700 gene fusions are now known in malignant diseases [1]. Each fusion gene has specific roles and functions in the genesis and progression of various cancers, and not all the genes account for the driver mutations that are essential to cancer growth and survival. Thus, not all the fusion genes are direct treatment targets for molecular therapies, like *BCR-ABL* and *ALK*. Nevertheless, these gene fusions provide important clues to both the classification of malignant diseases based on macro- or microscopic findings and the clarification of the disease mechanisms. This is also true for soft tissue sarcomas (STS).

In this review, we summarize the current understanding of chromosomal translocations in STS, regarding both clinical features and new findings from basic science. We discuss the new molecular therapies targeting these translocations and the related mutations that are being studied or tested in ongoing trials.

### 1.2. Chromosomal Translocations in Soft Tissue Sarcoma

STS is a rare tumor accounting for only 5% of all malignant tumors, but there are diverse heterogenic components among the subtypes of STS. There are more than 50 pathological subtypes of STS in the current World Health Organization (WHO) classification [7]. The progress that has been made in pathological technology and the accumulation of pathological and clinical data have enabled precise diagnoses and classifications. The identification of translocations is one example of this. Approximately 300 gene fusions are known in mesenchymal tumors, and ~20% of the STS subtypes have chromosomal translocations; these are called translocation-related sarcomas (TRSs) [1]. The major TRSs and their specific translocations and fusion genes are shown below (Table 1, Figure 1).

The clinical significance of chromosomal translocations as treatment targets in STS has been unclear in most TRSs, though some of the TRSs are thought to act as regulators of transcription. Chromosomal translocations are important factors to be considered in pathological diagnoses, but they were not regarded as treatment targets until recently.

The antineoplastic agent, trabectedin, which was initially approved in Europe for the treatment of STS as an alkylating agent, was reported to show more clinical benefits against translocation-related sarcomas than against those without translocations by retrospective data after its approval [8]. Based on those data, clinical trials of trabectedin as a treatment for TRS were performed and showed clinical benefits [9,10]. Trabectedin is thought to have cytotoxic effects by joining minor grooves of DNA in tumor cells, and these effects may be augmented in tumor cells with the instability of DNA repairs, which would account for the finding that trabectedin has a greater effect on TRSs compared to STS without translocation [11]. The results of other retrospective analyses suggest a relationship between the response to trabectedin and its DNA repair function—including mutations in BRCA (breast cancer susceptibility gene), which are well known as risk factors of hereditary cancers [12,13,14]. Other antitumor agents targeting the deficiency of DNA repair in STS are under investigation in clinical trials [15,16].

However, there are differences in trabectedin’s effects on TRSs among the different pathological subtypes. In some specific histologies, such as Ewing sarcoma and myxoid liposarcoma, pharmacologic effects of trabectedin acting directly on fusion genes were suggested by the results of preclinical studies [17].

The above-noted findings concerning trabectedin indicate that each characteristic and function of fusion genes should be considered when the optimization of targeted therapies to TRS is desired, along with the detection of the presence of chromosomal translocation(s). In the next section, we describe each TRS: Their fusion proteins, clinical characteristics, and candidates for molecular-targeted therapies.

## 2. Subtypes of Translocation-Related Sarcomas: Clinical Significance

### 2.1. The Ewing Sarcoma Family of Tumors

#### 2.1.1. Ewing Sarcoma

Ewing sarcoma is a representative small round cell sarcoma that originates from both bone and soft tissue and occurs mainly in children, adolescents, and young adults [18]. The chromosomal translocation observed most often in Ewing sarcoma is t(11;22)(q24;q12), which brings the fusion gene of *EWSR1-FLI1*. Other subtypes of Ewing family sarcomas usually have fusion proteins derived from the translocations of *EWSR1* (Ewing sarcoma breakpoint region 1, also known as EWS). There are many sarcomas other than Ewing sarcoma that also have EWRS1-related fusion proteins, e.g., clear cell sarcoma, desmoplastic round small-cell tumor, and extraskeletal myxoid chondrosarcoma.

The relationship between the differences in fusion genes and the differences in prognoses in Ewing sarcoma is controversial. In some retrospective analyses, Ewing sarcoma patients with *EWSR1-FLI1* showed better prognoses than patients with other translocations [19,20], but in a comparison of Ewing sarcoma patients’ survival in prospective trials, a significantly favorable survival of patients with *EWSR1-FLI1* was not confirmed [21]. The functions of *EWSR1-FLI1*, which was identified as an oncogene, include the regulation of the expression of insulin-like growth factor 1 (IGF1) and its relationship with the poly-ADP-ribose-polymerase (PARP) pathway; the overexpression of IGF1 receptor (IGF1R) increases cell survival [22]. *EWSR1-FLI1* is also known as a regulator of the *NR0B1* gene, which plays a transcriptional role in tumor genesis [23].

Ewing sarcomas are usually highly sensitive to cytotoxic antitumor agents, and thus intensive chemotherapy with a combination of cytotoxic drugs, including cyclophosphamide, doxorubicin, dactomycin, vincristine, ifosfamide, and etoposide, is included in the standard treatment strategy. An approximate 70% five-year event-free survival rate was reported in non-metastatic Ewing sarcoma patients [24]. Accelerations of the dose intensity of cytotoxic drugs were attempted to improve patient outcomes, but the clinical benefits of these accelerations were limited [25,26]. High-dose chemotherapy followed by autologous cell transplantation for localized high-risk patients could be superior to standard chemotherapy, but in such an intensive treatment setting, a high rate of adverse events (both acute and late) would be observed [27]. When Ewing sarcomas relapse, they progress quickly and are often lethal. New treatment strategies for recurrent and/or metastatic Ewing sarcoma patients are needed. Even for curable patients, most of whom are <20 years old, molecular-targeted therapy with a low risk of late toxicities would be welcome.

As described above in Section 1.2, chromosomal translocation itself could be a treatment target of trabectedin, and the specific translocation of Ewing sarcoma could be the target of trabectedin. In a preclinical study, trabectedin inhibited *NR0B1* expression by suppressing the activity of *EWSR1-FLI1* [28]. In a clinical trial of trabectedin for TRSs, a partial response to recurrent/metastatic Ewing sarcoma was observed [29].

The IGF1R pathway is deregulated by the *EWSR1-FLI1* translocation, which makes this pathway a potential target for therapy. IGF1R-targeted therapies for many solid tumors (including Ewing sarcomas) have been investigated. For example, figitumumab is one of the IGF1R inhibitors subjected to clinical trials of its efficacy and safety for patients with Ewing sarcomas. In early-phase clinical trials, objective responses to the figitumumab treatment of Ewing sarcomas were observed [30,31]. However, a clinical trial of figitumumab for non-small cell lung cancer was terminated due to negative results [32].

In a phase 2 clinical trial, ganitumab, another IGF1R-targeted drug, also produced objective responses to Ewing sarcoma and other tumors in the Ewing sarcoma family [33]. Although a clinical trial of ganitumab for pancreatic cancer was terminated due to futility [34], ganitumab monotherapy and in combination with other agents for Ewing sarcoma and other small round cell sarcomas are being developed, and a phase three trial is currently ongoing [35,36,37].

#### 2.1.2. Ewing Sarcoma-like Small Blue Round Cell Tumors

There are small blue round cell sarcomas in the Ewing sarcoma family in which *EWSR1-* or other known gene fusions have been observed. Translocations with *CIC*, present at chromosome 19, were identified in some of these tumors, and *CIC-*arranged sarcoma is defined as a genetic variant of Ewing-like sarcoma [38]. The reported partner genes of *CIC* are *DUX4* and *FOXO4*, the latter of which is a member of the O subgroup of the forkhead box (FOX) family, as described below in the discussion of alveolar rhabdomyosarcomas (Section 2.2). Due to the rarity of these tumors, the clinical and epidemiologic information concerning STSs with *CIC-*arranged translocation is limited, but these tumors are relatively resistant to the conventional chemotherapies used for Ewing sarcoma [39]. The presence of treatment targets around these translocations has not been investigated.

#### 2.1.3. Desmoplastic Small Round Cell Tumors (DSRCTs)

A standard treatment strategy for desmoplastic small round cell tumors (DSRCTs), which have the translocation t(11;12)(p13;q12), has not been established. DSRCT is an example of small round cell tumors (i.e., it is a “Ewing-family member”). Multimodal therapy, including combination chemotherapy similar to the strategy for Ewing sarcomas, is often performed for DSRCTs, but the prognosis remains poor [40,41]. As salvage systemic chemotherapy, the antiangiogenic tyrosine kinase inhibitor, pazopanib, produced some clinical responses to DSRCTs, including complete response (CR) in a small retrospective study, but the overall response rate was limited to <10% [42].

As DSRCTs harbor *EWSR1-*related translocations, they could be examined in clinical trials that target EWSR1 or its surrounding signaling molecules along with other Ewing-family tumors [30,33].

### 2.2. Alveolar Rhabdomyosarcomas

Like Ewing sarcoma, rhabodomyosarcoma is included in the group of small cell round tumors, and it has several subtypes: Embryonal, alveolar, and pleomorphic rhabodomyosarcomas. The embryonal type arises in infants and children, and as the age of patients increases, the percentage of the embryonal type declines. Conversely, the percentages of the alveolar and pleomorphic types rise in adolescents and older patients.

Chromosomal translocation is observed particularly in alveolar rhabdomyosarcoma, including two patterns of translocations: t(2;13)(q35;q14) with the *PAX3-FOXO1* fusion gene, and t(1;13)(p36;q14) with the *PAX7-FOXO1* fusion gene. The prevalence of translocation with *PAX3-FOXO1* is higher than that with PAX7-FOXO1 [7].

*FOXO1*, formerly called *FKHR* (forkhead in rhabdomyosarcoma), is a member of the forkhead box (FOX) family [42]. The O subgroup of the FOX family includes four members (*FOXO1*, *FOXO3*, *FOXO4*, and *FOXO6*), of which *FOXO4* was mentioned above in Section 2.1.2. *FOXO* factors are considered tumor suppressors that are inactivated by the phosphatidylinositol 3-kinase (PI3K)-AKT pathway, regulated by several micro-RNAs. *FOXO-*related fusion proteins contribute to oncogenesis, but they are not considered mandatory mutations; this is based on preclinical data that conditional knock-in Pax3-Foxo1 mice did not develop tumors without other concomitant mutations [43].

The predictive value of *FOXO1-*related translocations in alveolar rhabdomyosarcoma has been investigated. It was suggested that the presence of detectable fusion genes involving *FOXO1* could be the predictive factor of poor prognosis in lymph node-positive alveolar rhabdomyosarcoma [44]. In terms of the differences in translocations, for both non-metastatic and metastatic alveolar rhabdomyosarcoma patients, the translocation of *PAX3-FOXO1* was suggested to result in shorter survival than *PAX7-FOXO1* translocation [45,46,47].

Like other small round cell sarcomas, rhabdomyosarcoma is highly sensitive to cytotoxic chemotherapy; the combination chemotherapy of vincristine, dactinomycin, and cyclophosphamide is a standard treatment regimen [48]. Unlike other STS, however, it seems that doxorubicin does not contribute to an improvement in the overall survival of rhabdomyosarcoma patients [49]. For low- or intermediate-risk rhabdomyosarcoma patients (most of which have been pediatric patients with embryonal-type tumors), a high cure rate would be expected for the current standard treatment; there is a clinical trial of low-intent chemotherapy being used to avoid late toxicities [50]. In adolescents and older patients, most of whom have had alveolar- or pleomorphic-type rhabdomyosarcomas, the prognoses would be poor [51]. Adding other cytotoxic agents and empowering the dose intensity have been tried to improve clinical efficacy in a single-arm clinical trial, but their evidence to improve OS has not been sufficient yet [52].

*PAX3-FOXO1*, the fusion gene that is a risk factor for poor prognosis, acts as a super-enhancer dependent on the BET bromodomain protein, BRD4 (bromodomain-containing protein 4), which suggests *BRD4* as a treatment target for alveolar rhabdomyosarcoma patients with the *PAX3-FOXO1* translocation [53]. In preclinical models, antitumor responses of pediatric sarcomas (including rhabdomyosarcoma) to a BET inhibitor were observed [54]. Now that clinical trials of BET inhibitors have started, clinical data of BET inhibitors used to treat rhabdomyosarcoma are awaited [55]. The effects of interleukin (IL)-24 for directly targeting *PAX3-FOXO1* inhibition are also under investigation in a preclinical study [56].

*ALK-*related mutations, which are well known in lung cancer as noted in the Introduction, are also seen in rhabdomyosarcoma, especially in the alveolar type; an *ALK* gene copy number gain is detected in the vast majority of alveolar rhabdomyosarcomas [57]. However, these mutations are genetic aberrations at the mRNA level in the specific domain of the *ALK* gene, and they are different from the *ALK-*related translocations observed in NSCLCs and inflammatory myofibroblastic tumors as we will discuss below in Section 2.7. These differences might be the reason why the *ALK* inhibitors did not show clinically meaningful effectiveness against alveolar rhabdomyosarcomas [58,59].

### 2.3. Alveolar Soft Part Sarcoma (ASPS)

A rare subtype of STS is alveolar soft part sarcoma (ASPS), which accounts for <1% of all STS and occurs mostly at adolescent and young adult (AYA) ages [7]. An unbalanced chromosomal translocation, t(X;17)(p11;q25), is characteristic of ASPS, and it is also found in pediatric papillary renal cell cancers [60,61]. The translocation t(X;17)(p11;q25) includes the fusion of *ASPL-TFE3*, which brings the *TFE3* DNA-binding domain, implicating transcriptional deregulation in the pathogenesis of ASPS [62]. The detection of *TFE3* by immunohistochemistry is useful for the pathological diagnosis of ASPS [63].

It is possible to cure ASPS in a localized setting by surgical resection, but a high proportion of ASPSs have multiple metastases in lungs, and the long-term prognoses of patients with ASPS recurrence and/or metastases are still poor [64,65]. The tendency of ASPS to metastasize might be explained by the activities observed in preclinical models that introduced *ASPL-TFE3* translocation [66].

Conventional cytotoxic chemotherapies for recurrent and/or metastatic ASPS have seldom shown clinical benefits. In a series of ASPS patients treated with trabectedin, modest benefits of disease stabilization were shown; however, objective responses were rarely observed [67]. In contrast, an anti-angiogenetic agent is known to be effective. Cediranib, an oral tyrosine kinase inhibitor that inhibits the vascular endothelial growth factor receptors (VEGFR)-1, -2, and -3, produced a 35% rate of objective responses among patients with metastatic ASPS in a phase 2 trial [68]. Other tyrosine kinase inhibitors, such as sunitinib, have also shown clinical benefits against ASPS [69,70]. The relationship between clinical responses and ASPS-specific translocations is also being investigated in clinical trials; MET mutation is an example [71]. The tyrosine kinase inhibitor, pazopanib, is currently the only tyrosine kinase inhibitor approved for the treatment of STS, and it is frequently used in clinical practice [72].

There are case reports of ASPS patients who responded to immune checkpoint inhibitors, and these responses were suggested to have occurred because of the molecular mismatch-repair deficiency signatures that are demonstrated in ASPS tumor cells [73,74]. DNA mismatch-repair deficiency is now known as an important predictive factor of clinical responses to immune checkpoint inhibitors [75], and further information about the frequency of DNA mismatch-repair deficiency in STS, including ASPS, is needed.

### 2.4. Synovial Sarcoma

Synovial sarcoma is characterized by the translocation t(X;18)(p11;q11), which fuses *SS18* (SYT) in chromosome 18 and *SSX1*, SSX2, or *SSX4* (rarely) in chromosome X [7]. It occurs mostly in adolescents and young adults (i.e., AYA), but there is no difference in patient numbers in terms of sex [76]. The relationship between prognoses and fusion genes is controversial. Some studies suggest that synovial sarcoma patients with the fusion gene, *SS18-SSX2*, have better prognoses than those with the fusion gene, *SS18-SSX1* [77,78], but the difference in prognoses was not reproduced in other studies [79,80].

Synovial sarcoma is known to be more sensitive than other sarcomas to ifosfamide, an alkylating agent. For patients with a recurrence of metastatic synovial sarcoma, the use of ifosfamide would be preferred than for another histology of STS, though the appropriate doses and cycles are not established [81]. Pazopanib has also shown relatively better clinical responses among the subsets of STS [82].

*SSX-SS18* fusion genes are thought to be related to an alteration in chromatin remodeling and to the expression of a cancer immunity antigen known as New York esophageal squamous cell carcinoma 1 (NY-ESO-1) [76]. In epigenetic disorders with *SSX-SS18* fusion gene origins, a candidate treatment target in synovial sarcoma shown by preclinical data is an enhancer of zeste homologue 2 (EZH2) [83]. Clinical trials of EZH2 inhibitors have begun, along with investigations of these inhibitors for STS, including synovial sarcoma [84,85]. NY-ESO-1 is one of the cancer-testis antigens, and its tumor expression is thought to correlate with the immune response in many malignant diseases [86].

Despite the high expression of NY-ESO-1 and expected responses to immunotherapy, immune checkpoint inhibitors, including anti-PD-1 (programmed death-1) or -PD-L1 (programmed death-ligand 1) antibodies, did not produce meaningful clinical responses in patients with synovial sarcomas [87,88,89]. Nevertheless, immunotherapy directly targeting NY-ESO-1 could be promising in approaches, such as the engineering of T-cell therapy in which engineered T cells that recognize NY-ESO-1 are infused to patients [90]. In a prospective clinical trial, engineered T-cell therapy resulted in 50% confirmed clinical responses, and the responses persisted [91].

### 2.5. Myxoid Liposarcoma

Liposarcoma is one of the major pathological subtypes of STS, and its subtypes include well-differentiated liposarcoma, dedifferentiated liposarcoma, and myxoid liposarcoma. Of these, myxoid liposarcoma accounts for 20%–30% of liposarcomas and originates mostly in the lower extremities [92]. Myxoid liposarcoma is characterized by a specific translocation, t(12;16)(q13;p11), which brings the *FUS-CHOP* fusion gene; rarely, another translocation of t(11;22)(q13;q12) with the *EWSR1-CHOP* fusion gene has been observed [93]. *FUS* is thought to be involved in transcriptional activation [94]. These fusion genes act as abnormal transcription factors, and they generate tumor progression.

Perhaps by the differences of mechanisms in tumor genesis, the sensitivity of myxoid liposarcoma to antitumor drugs differs from those of the other liposarcoma subtypes. For example, eribulin, a microtubule targeting cytotoxic drugs, is known to prolong overall survival in patients with liposarcoma, but the clinical benefits of eribulin were more significant in dedifferentiated liposarcoma compared to myxoid liposarcoma [95]. Myxoid liposarcomas were more sensitive than other liposarcomas to trabectedin [96]. In a clinical trial of neoadjuvant chemotherapy, the myxoid liposarcoma cohort treated with trabectedin showed non-inferior responses compared to the cohort treated with standard neoadjuvant chemotherapy (anthracycline and ifosfamide), whereas most of the other histology-tailored chemotherapies have failed to show superiority or non-inferiority to standard chemotherapy [97]. The high response of myxoid liposarcoma to trabectedin was the trigger of the discovery that trabectedin is effective against TRSs [98]. In myxoid liposarcoma with the *FUS-CHOP* fusion gene, the pharmacological mechanism of trabectedin is to block the trans-activating chimera, resulting in the detachment of the *FUS-CHOP* chimera from targeted promoters [99].

### 2.6. Clear Cell Sarcoma (CCS)

Clear cell sarcoma (CCS) resembles malignant melanoma in some pathological features, and thus CCS has sometimes been referred to as “melanoma of soft parts.” The key distinctive factor of CCS and melanoma is the t(12;22)(q13;q12) translocation, which is almost observed in CCS, but not in malignant melanoma [100,101]. The fusion gene includes *EWSR1* in chromosome 22 (as noted in the earlier section about Ewing sarcomas), and thus IGF1R expression might be a clue for distinguishing CCS from melanoma [102]. *BRAF* and *NRAS* mutations (which are important treatment targets in malignant melanoma) are not usually observed in CCS, although there is an exceptional CCS case that harbored *BRAF* mutation and responded to *BRAF-*targeted therapy [103].

Immune checkpoint inhibitors, including anti-PD-1, anti-PD-L1, and anti-CTLA-4 (cytotoxic T-lymphocyte-associated protein 4) antibodies, have become the standard therapy for malignant melanoma [104]. Whether immune checkpoint inhibitors are also promising for CCS remains to be established, but there is a clinical report that anti-PD-1 therapy produced a clinical response among CCS patients [105].

### 2.7. Inflammatory Myofibroblastic Tumor (IMT)

Inflammatory myofibroblastic tumor (IMT) is an extremely rare type of STS. The most well-known pathological feature of IMT is *ALK*, mentioned in the Introduction as a driver gene in NSCLC. In over 50% of IMT patients, *ALK* overexpression is detected by immunohistochemistry, and *ALK-*related translocations are also detected; there are variations of partner fusion genes, i.e., *TPM3*, *TPM4*, *CLTC*, *RANBP2*, and *ATIC* [7]. However, IMTs that do not harbor *ALK-*related translocations have been described, and these *ALK-*negative IMTs have other specific translocations, such as *ROS1*, *PDGFR*, *RET*, or *NTRK3* instead [106,107,108].

The clinical features of IMT are, as its name implies, characterized by inflammatory symptoms: Fever, pain, and high inflammatory markers. Like *ALK-*positive NSCLC, cranial metastases are often observed in IMT patients [109,110]. There are some case reports that anti-inflammatory treatments by corticosteroids or non-steroidal anti-inflammatory drugs (NSAIDs) were effective for tumor reduction as well as symptom palliation in IMT patients [111,112], but IMTs are usually resistant to the traditional cytotoxic chemotherapies approved for STS. As the pathological and molecular characteristics of IMT had been unveiled, clinical responses to ALK inhibitors were expected for IMT patients with *ALK-*related translocation. Following the first case report of IMT patients who showed durable responses to the first ALK inhibitor identified, i.e., crizotinib [113], there have been many reports of IMT patients who were successfully treated with ALK inhibitors [114,115,116]. Prospective clinical trials of crizotinib treatment for IMT confirmed the results of the case reports; in a phase two trial, a complete response was observed in 36% (five of 14) of patients with IMT [117,118].

### 2.8. Infantile Fibrosarcoma

The translocation of t(12;15)(p13;q25), resulting in the fusion gene *ETV6-NTRK3*, is related to infantile fibrosarcoma, which occurs in infants. An infantile fibrosarcoma originating from the kidney is called congenital mesoblastic nephroma [119]. *ETV6-NTRK3* fusion gene products activate signaling cascades, including the ras protein (RAS) and phosphatidylinositol 3-kinase-protein kinase pathways [120]. It is possible to cure an infantile fibrosarcoma with a complete surgical resection, but these tumors tend to progress rapidly, and thus some cases are inoperable [121].

*NTRK-*related fusion genes, including *ETV6-NTRK3*, were recently observed in many solid tumors although at a low rate, and these genes could become new treatment targets [122,123]. NTRK inhibitors, such as larotrectinib (LOXO-101) and entrectinib, are under investigation, and in early-phase clinical trials, they have produced excellent responses in cancers with *NTRK-*related translocations [124,125]. Objective responses of infantile fibrosarcomas to NTRK inhibitors are also reported [126,127]. With dose modifications based on age and/or body weight, a determination of the safety of NTRK inhibitors for infants and young children over a long follow-up for late toxicities is needed. In parallel to the confirmation of the safety and efficacy of the existing NTRK inhibitors, investigations of next-generation NTRK inhibitors are underway, and acquired resistance to NTRK inhibitors is being addressed [128,129].

### 2.9. Other TRS

Other than TRS described above, STS known to have their specific translocations includes myxoinflammatory fibroblastic sarcoma with t(1;10)(p22-31;q24-25), low-grade fibromyxoid sarcoma with t(7;16)(q33;p11), extraskeletal myxoid chondrosarcoma with t(9;22)(q22;q12), and, moreover, there are many benign tumors with specific translocations [7]. To these TRS, however, new treatment strategies targeted to their specific translocations have not been established. 

## 3. Translocation Analysis Technology: The Detection and Induction of Fusion Genes

Until recently, the chromosomal translocations discussed above have been detected by fluorescence in situ hybridization (FISH) or polymerase chain reaction (PCR). These methods have shown high sensitivity and specificity for detecting translocations that are already known, and they are widely used in clinical practice and clinical trials [130]. However, each FISH and PCR evaluates a specific target — a mutation, fusion gene, or protein — and thus FISH and PCR are limited for determining the presence or absence of many mutations in tumor tissues concomitantly and comprehensively. For example, *NTRK-*related fusion genes (referred to above Section 2.8 about infantile fibrosarcoma) are observed in many malignant diseases, but these mutations are present in each cancer at low rates. Without the existence of *NTRK-*targeted therapies and their high clinical responses, the translocations would usually not be detected. In fact, there is a clinical report of a patient who was not initially diagnosed with TRS, but was eventually shown to harbor a *NTRK-*related fusion gene and responded to an NTRK inhibitor [131]. As new treatable targets are established and new targeted therapies are designed, many mutations, including translocations, must be evaluated concomitantly to provide optimal precision medicine.

The new technologies that can be used to address these tasks are large-scale sequencing, i.e., so-called next-generation sequencing (NGS) [132,133]. For the evaluation of the functions of fusion genes and proteins, transcriptome analyses are considered to be more effective than genome sequencing in STS investigations [134]. The advances in these technologies have recently revealed new translocations. For example, a comprehensive analysis of the tumor genomics demonstrated that pediatric undifferentiated sarcomas, which are not formally recognized as TRSs, harbor clinically relevant oncogenic fusion genes [135]. In another study, integrative genomic and transcriptome analyses unveiled some translocations in leiomyosarcoma, which was not previously regarded as a TRS [136]. In the near future, the definition of TRS might be changed based on new findings obtained with these sequencing technologies.

Although comprehensive genome and transcriptome analyses may clarify known or unknown translocations, the targeted therapy drugs now available or under investigation do not yet cover all the translocations. In preclinical investigations, gene editing technologies have been greatly enhanced by CRISPER-Cas9 (clustered regularly interspaced short palindromic repeats-CRISPR-associated 9) [137]. This new technology has enabled the direct induction of oncogenic translocations, as observed in TRS. Investigators could thus use tumor tissues with rare mutations, including translocations, and lower the thresholds for developing new drugs to treat patients with such mutations [138,139].

## Figures and Tables

**Figure 1 ijms-19-03784-f001:**
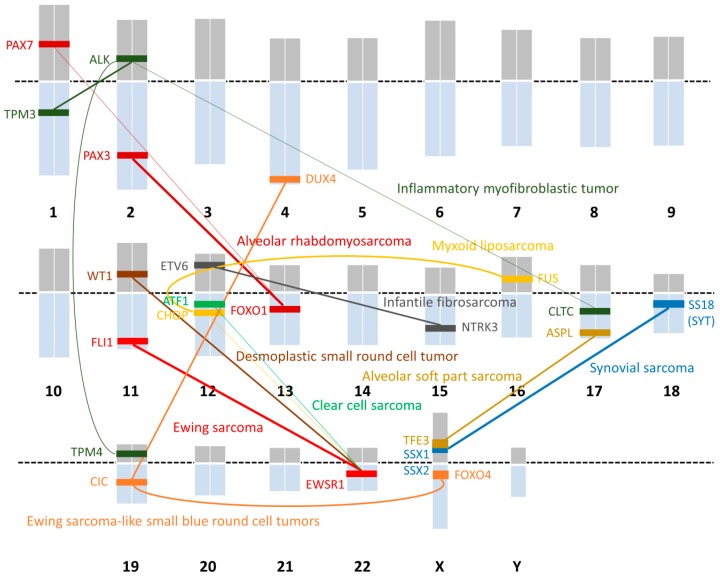
Chromosomal translocations in STS and their locations. Abbreviations: ALK; anaplastic lymphoma kinase, DUX4; double-homeobox, FOXO; forkhead box transcription factor O, NTRK; neurotrophic tyrosine kinase, TFE3; transcription factor E3.

**Table 1 ijms-19-03784-t001:** Chromosomal translocations in soft tissue sarcoma (STS).

Histological Subtype	Chromosomal Translocation	Fusion Gene
Ewing sarcoma	t(11;22)(q24;q12)	*EWSR1-FLI1*
	t(21;22)(q22;q12)	*EWSR1-ERG*
	t(7;22)(q22;q12)	*EWSR1-ETV1*
	t(17;22)(q12;q12)	*EWSR1-ETV4*
	t(2;22)(q33;q12)	*EWSR1-FEV*
	t(16;21)(p11;q22)	*FUS-ERG*
Ewing sarcoma-like small blue round cell tumor	t(4;19)(q35;q13)	*CIC-DUX4*
	t(X;19)(q13;q13)	*CIC-FOXO4*
Desmoplastic small round cell tumor	t(11;22)(p13;q12)	*EWSR1-WT1*
Alveolar rhabdomyosarcoma	t(2;13)(q35;q14)	*PAX3-FOXO1*
	t(1;13)(p36;q14)	*PAX7-FOXO1*
Alveolar soft part sarcoma	t(X;17)(p11;q25)	*ASPL-TFE3*
Synovial sarcoma	t(X;18)(p11;q11)	*SS18-SSX1*
	t(X;18)(p11;q11)	*SS18-SSX2*
	t(X;18)(p11;q11)	*SS18-SSX4*
Myxoid liposarcoma	t(12;16)(q13;p11)	*FUS-CHOP*
	t(12;22)(q13;q12)	*EWSR1-CHOP*
Clear cell sarcoma	t(12;22)(q13;q12)	*EWSR1-ATF1*
Inflammatory myofibroblastic tumor	t(1;2)(q21;p23)	*TMP3-ALK*
	t(2;19)(p23;p13)	*TMP4-ALK*
	t(2;17)(p23;q23)	*CLTC-ALK*
Infantile fibrosarcoma	t(12;15)(p13;q25)	*ETV6-NTRK3*

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
