# Peer review of "Translocation-Related Sarcomas"

_ijms, 2018, doi:10.3390/ijms19123784_

Round 1

Reviewer 1 Report

This is a well written and comprehensive review of translocation-related sarcomas.  A few minor edits/comments for consideration:

Authors use STS or STSs for their abbreviation.  Recommend keeping STS for consistency throughout article. 

Line 120- considering adding "often" prior to lethal as some recurrent ES are salvageable.

Line 139- change to currently ongoing.  There is a phase 3 trial of ganitumab ongoing (NCT02306161).

Line 188 change from the to a since there are other recommended upfront therapies for RMS.

Line 195-196- Would change.  The addition of more cytotoxic agents have not lead to changes in OS for ARMS.

Line 253= would change.  Many have received upfront ifosfamide as therapy.  Also many synovial sarcomas are chemo-resistant and thus stating anything is mandatory is not recommended.

For the paragraph starting in Line 279- Could authors clarify or make statement as to why they think there are differences in these responses to varying treatments based histology. 

Author Response

Thank you for reviewing our article. We corrected the article under your advices as below:

・We changed all "STSs" abbreviations to "STS" for consisitency.
・In Line 120- we added  "often" prior to lethal as some recurrent ES are salvageable.
・In Line 139-  we changed to currently ongoing.  A phase 3 trial of ganitumab ongoing was added to the reference.
・In Line 188 we changed from "the" to "a" since there are other recommended upfront therapies for RMS.
・We corrected Line 195-196- that the evidence of addition of more cytotoxic agents have not been established.
・We changed Line 253 that ifosfamide was "preferred" than "mandatory".
・For the paragraph starting in Line 279- we  added "Perhaps by the differences of mechanisms in tumor genesis", as the consideration why they think there are differences in these responses to varying treatments based histology.

Reviewer 2 Report

In the present article the authors review chromosomal translocations in soft tissue sarcomas. The focus of the review is not original and the content is not complete.

To make the manuscript attractive to readers the authors might include all sarcoma types (bone and soft tissue) and accordingly add information of all of them (grouped following WHO's classification).

Author Response

Tnak you for reviewing our arcilce. Under your advices, we added the section "2.9 Other TRS" and referred to the other TRS described in the WHO classification.

Round 2

Reviewer 2 Report

The new section "2.9 Other TRS" is very short and the review would gain interest if the authors would add some additional information on these tumor types. However, in my opinion this version of their manuscript is acceptable for its publication in IJMS.